# Semantics-and-Primitives-Guided Indoor 3D Reconstruction from Point Clouds

**Tengfei Wang, Qingdong Wang *, Haibin Ai and Li Zhang**

Institute of Photogrammetry and Remote Sensing, Chinese Academy of Surveying and Mapping (CASM), Beijing 100036, China
* Correspondence: wangqd@casm.ac.cn

**Abstract:** The automatic 3D reconstruction of indoor scenes is of great significance in the application of 3D-scene understanding. The existing methods have poor resilience to the incomplete and noisy point cloud, which leads to low-quality results and tedious post-processing. Therefore, the objective of this work is to automatically reconstruct indoor scenes from an incomplete and noisy point-cloud base on semantics and primitives. In this paper, we propose a semantics-and-primitives-guided indoor 3D reconstruction method. Firstly, a local, fully connected graph neural network is designed for semantic segmentation. Secondly, based on the enumerable features of indoor scenes, a primitive-based reconstruction method is proposed, which retrieves the most similar model in a 3D-ESF indoor model library by using ESF descriptors and semantic labels. Finally, a coarse-to-fine registration method is proposed to register the model into the scene. The results indicate that our method can achieve high-quality results while remaining better resilience to the incompleteness and noise of point cloud. It is concluded that the proposed method is practical and is able to automatically reconstruct the indoor scene from the point cloud with incompleteness and noise.

**Keywords:** 3D reconstruction; semantic segmentation; indoor scene; point cloud; deep learning

## 1. Introduction

Indoor 3D reconstruction has achieved great success in indoor navigation, augmented reality and robots. Compared with conventional 3D geometric reconstruction, 3D semantic reconstruction has a broader range applications, such as 3D-scene understanding and analysis. However, the complexity and diversity of indoor scenes make it difficult to recover high-quality 3D shapes and accurate semantic information automatically, especially for unstructured objects, such as furniture, as the incompleteness and noise in point clouds seriously degrade the robustness of the existing methods. Therefore, it is still a challenging task to achieve automated and high-quality indoor 3D semantic reconstruction.

In recent years, many works have studied how to reconstruct the geometry and semantics of indoor scenes. Most of the related methods can be classified into four types: (1) surface-reconstruction-based methods, (2) geometric-prior-based methods, (3) primitive-based methods, and (4) deep-learning based methods. The surface-reconstruction-based methods are mainly used for indoor navigation, and the data source is usually captured by a depth-camera device. Izadi proposed KinectFusion [1], which uses the truncated signed distance function (TSDF) to describe the geometric surface of the object, and iteratively updates the TSDF value by the depth data. Based on this work, Whelan et al. [2] proposed Kintinuous, which improves the limitation of KinectFusion by introducing volume shifting and the loop-closure constraint. To further improve the accuracy, Whelan et al. [3] proposed the ElasticFusion method, which improves the accuracy of reconstruction and pose estimation by replacing TSDF with Surfel. The surface-reconstruction-based methods focus more on geometric information and incur high computational costs. The quality of the reconstructed results is also reliant on the quality of the data source. Moreover, the lack of semantic information limits the extension of these methods to other fields.

The geometric-prior-based method is mainly concerned with incorporating geometric-prior knowledge, such as point normal, spatial distance, and adjacency, vertical and horizontal, into the feature-extraction algorithm. Jung et al. [4] applied the RANSAC algorithm for plane detection and performed contour extraction. Wang et al. [5] used the region-growing algorithm to detect and classify building-plane and other components, and the Bayesian sampling consensus [6] method was also applied to 3D reconstruction. Such methods can reconstruct the geometry while preserving the corresponding semantic information, but they are only reliable for some simple structures (walls, floors, ceilings, etc.).

The primitive-based methods generally use the shape descriptors and shape primitives to decompose the object into several shape primitives, and reconstruct it through the composition of these shape primitives. Florent et al. [7] proposed a general-purpose parametric method to retrieve and fit the corresponding model from the 3D model library. Nan et al. [8] retrieved the most similar model from the 3D-model library based on geometric and symmetry descriptors. Xu et al. [9] took into account the shape style and proposed a style–content separation to create the novel instance from the shape set. In some other methods, parametric modeling is the key technology [10]. However, for objects with complex geometric structure, the analysis of 3D shapes and parameters is complicated and inaccurate; therefore, it also only reliable for objects with simple structures.

A few recent works focused on deep learning for point-cloud segmentation and 3D-shape prediction. For 3D-shape prediction, there are some typical works, such as Pixel2mesh [11], Mesh R-CNN [12], pixel2mesh++ [13] and DeepSDF [14]. Since these methods use the neural network to learn the implicit representation of 3D shapes or shape priors, they have the ability to reconstruct 3D geometry from sparse views and points. However, with regard to the incomplete point cloud, the predicted shape is not sufficiently accurate, especially for complex porous structures; therefore, there is still a large scope for improvement. By contrast, deep-learning-based point-cloud segmentation has achieved great progress. The related methods can be categorized into three types: voxel-based methods, point-based methods and GCN (Graph Convolution Network)-based methods. The voxel-based method is the extension of 2D CNNs. To apply standard CNNs on point clouds, generally, the point clouds need to be voxelized, after which the 3D convolution kernels can be applied; typical works include 3DCNN [15], VoxNet [16] and SEGCloud [17]. Although many researchers have proposed various improvements to the voxel-based method, the limitations, including information loss, large memory consumption and computational expensiveness, are still difficult to solve. In order to apply convolution on point clouds directly, C.R. Qi et al. proposed the point-based method, PointNet [18]. As a pioneering method, PointNet satisfies the sparsity, permutation and rotation invariance of point clouds. Based on this work, a series of improvemed methods, such as PointNet++ [19], PointSIFT [20] and PointCNN [21] were proposed. Meanwhile, with the development of graph neural networks [22], the GCN-based method [23] was proposed for semantic or instance segmentation. Unlike the point-based method, the GCN method takes into account not only the points, but also the edges in the neighborhood, thereby achieving better performance. Related methods, such as DGCNN [24], SPG [25] and RandLA-Net [26], obtained a distinct improvement.

Overall, although the existing methods have made great progress, there are still several challenges to be resolved:

(1) Low degree of automation. Most of the existing methods reconstruct high-quality 3D semantic models manually or semi-automatically.
(2) Insufficient semantic utilization. Existing modeling methods do not fully utilize the semantic information of point clouds, which contains the priors that can improve the resilience to the noise and incompleteness of point clouds.
(3) Loss of details in geometry and incomplete detailed characterization. Existing methods often do not fully describe details such as edges and corners, especially for complex shapes, which affects the visualization.

(4)     Oversized geometric data. The existing methods mostly generate triangular mesh from dense point clouds. The data size is much larger than that of the semantics, which is inefficient for visualization.

To resolve the issues mentioned above, in this paper, we propose a semantics-and-primitives-guided indoor 3D reconstruction method. In this method, a local fully connected graph-based neural network (LFCG-Net) was designed for point-cloud segmentation. To obtain the instanced point clouds, such as a chair and a desk, a clustering algorithm was used to further segment the results from the semantic segmentation. Next, the semantic labels and the instanced point clouds were used to retrieve the most similar template model in an ESF (Ensemble of Shape Functions)-descriptor-based [27] 3D-model library (3D-ESF indoor model library), which was constructed for this study. Finally, to reconstruct the scene rapidly, we applied a coarse-to-fine registration algorithm to register the model to the indoor scene. Additionally, in order to verify the effectiveness of our method, we conducted verification experiments, and the results showed that our method is practical and effective. The key contributions of our work are as follows:

(1)     We propose an efficient and accurate point-cloud semantic-segmentation network (LFCG-Net).
(2)     Based on the enumerable features of indoor scenes, we propose a 3D-ESF indoor-model library and an ESF-descriptor-based retrieval algorithm.
(3)     We propose a robust and coarse-to-fine registration algorithm to rapidly reconstruct the 3D scene from the incomplete point cloud.

## 2. Materials and Methods

As shown in Figure 1, our method includes the following steps:

(1)     Semantic segmentation. We fed the point cloud into our LFCG-Net to obtain the semantic labels of the point cloud;
(2)     Instantiating clustering. Based on the semantic segmentation result, an instantiating-clustering algorithm was applied to separate the point clusters into individual clusters.
(3)     Unstructured-objects reconstruction. We retrieved the similar candidate models in our 3D-ESF indoor model library with the semantic label and ESF descriptors. Next, we registered the model to the scene;
(4)     Structured-objects reconstruction. The plane-fitting algorithm was used to model objects such as walls, floors, ceilings, etc.

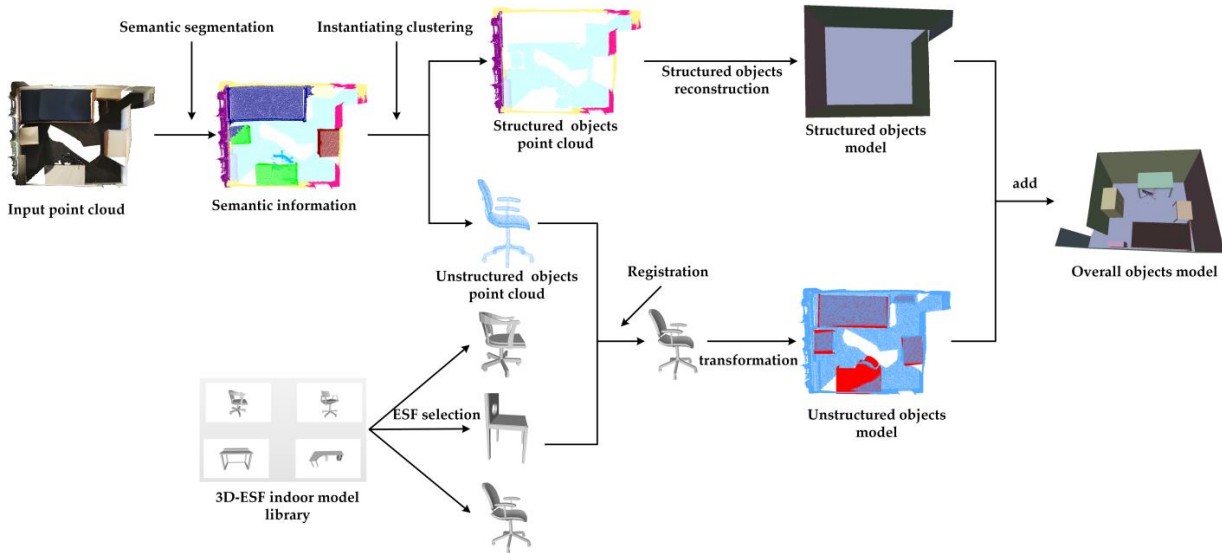

**Figure 1.** Flowchart of automatic semantic modeling of indoor point clouds.

### 2.1. Local Fully Connected Graph Network

Our network architecture is based on the Encoder–Decoder structure. There are two key modules in the network: the local fully connected graph spatial encoding module (LFCGSE) and the dual dilated residual block (DDRB). In the encoding stage, to improve computational efficiency, random downsampling was used to reduce the point density layer by layer. For the down-sampling results of each layer, we used LFCGSE to ensure that the local neighborhood features were fully encoded to solve the problem of incomplete semantic information description of local point clouds. Next, DDRB were used to extract high-dimensional features of point clouds and better local context information was obtained by continuously expanding the receptive field. In the decoding stage, we restored the original point-cloud density layer by layer through nearest-neighbor upsampling; subsequently, we used three fully connected layers to perform feature-dimension transformation. We used a dropout layer after the second fully connected layer to increase robustness with a dropout ratio of 0.5, and finally output the segmentation result. Considering the large variety and uneven distribution of objects in indoor point clouds, we used the weighted cross-entropy loss function for loss calculation. The network architecture is shown in Figure 2.

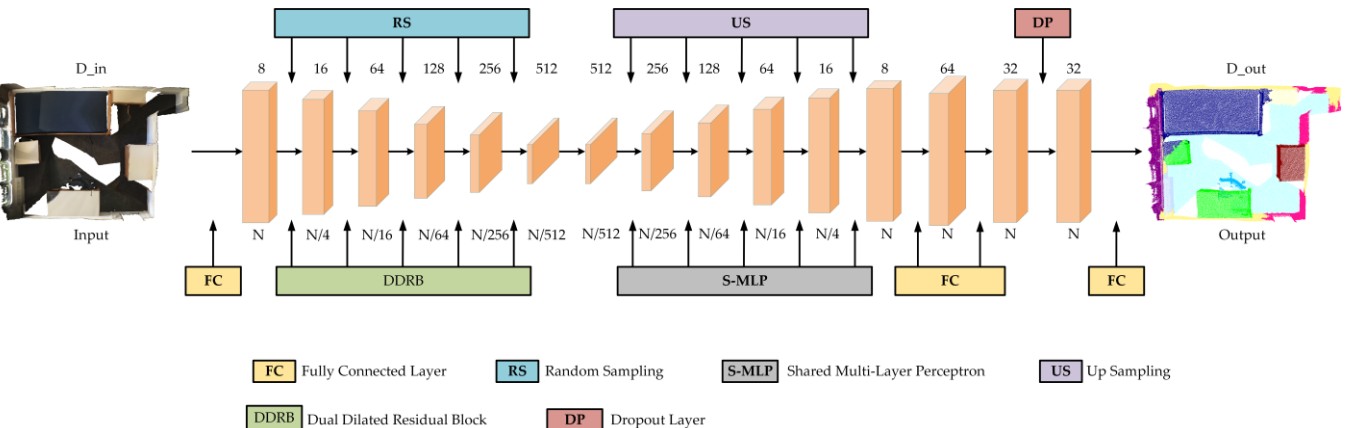

**Figure 2.** Network architecture. where D_in represents the input point-cloud dimension, D_out represents the output label dimension, the number above each layer structure in the figure is the dimension of each feature of this layer, and the number below is the number of features.

### 2.1.1. Local Fully Connected Graph Spatial-Encoding Module

Due to the sparsity and unordered nature of point clouds, it was not possible to use the convolution operation as 2D image, which occupies an inherent position among pixels. To address the problem, some methods about improving convolution on unordered points were proposed [18,19,21]. These methods were used to attempt to obtain the order-invariant point features by training a learnable transformation matrix. Recently, some related works focused on the neighborhood feature [24,28,29], which improves the convolution performance on unordered points by increasing more features, such as edges, points, density and Euclidean distance of neighborhood. The enhanced features can be considered as descriptions of the shape relation [29], which enhances the shape-awareness ability of the network and relieves the defect of unordered-point convolution.

I, there are also some related researches about shape relation description using traditional methods. Rusu et al. [30] proposed a local 3D descriptor, the Point Feature Histogram (PFH). The 3D descriptor improves the recognition accuracy by connecting each pair of nodes in the neighborhood graph, as the PFH descriptor is able to preserve more complete shape relations in the neighborhood compared to the existing neighborhood representation. Therefore, the representation of the local neighborhood features in point clouds is essential for improving the accuracy of segmentation or classification.

**The limitation of existing encoding methods**. The existing methods, which only take point $p$ and its neighbors $N(p)$ as the neighborhood, achieve some improvement. However,

the methods only describe the contextual shape relations between the center point and its neighbor points; the contextual shape relations among the neighbor points are ignored. That is to say, the contextual shape relations cannot be completely described in the existing neighborhood representations. Here, inspired by the PFH descriptor, we propose the local fully connected graph spatial encoding method for local spatial encoding, which encodes all pairs of points within the local neighborhood.

**Local fully connected graph spatial encoding**. We use *k-NN* to obtain the neighborhood of each point and construct the fully connected graph of neighborhood (Figure 3). Aside from encoding each edge and node in the neighborhood graph, the Euclidean distance of each pair of points is also encoded into the feature; for each neighbor point $p_i^k$, it is encoded as follows.

$$v_i^k = (p_i^k - p_i) \oplus (p_i^k - p_i^1) \oplus (p_i^k - p_i^2) \oplus \ldots \oplus (p_i^k - p_i^K) \tag{1}$$

$$u_i^k = \left\| \|p_i^k - p_i\| \oplus \|p_i^k - p_i^1\| \oplus \|p_i^k - p_i^2\| \oplus \ldots \oplus \|p_i^k - p_i^K\| \right\| \tag{2}$$

$$r_i^k = MLP\left(v_i^k \oplus u_i^k \oplus p_i\right) \tag{3}$$

where $p_i$ is the center point, $p_i^k$ is the neighbor points, $k$=1, 2,3 … $K$, $\oplus$ is the concatenation, $\|\cdot\|$ is the Euclidean distance, $r_i^k$ is the encoding feature of $p_i^k$.

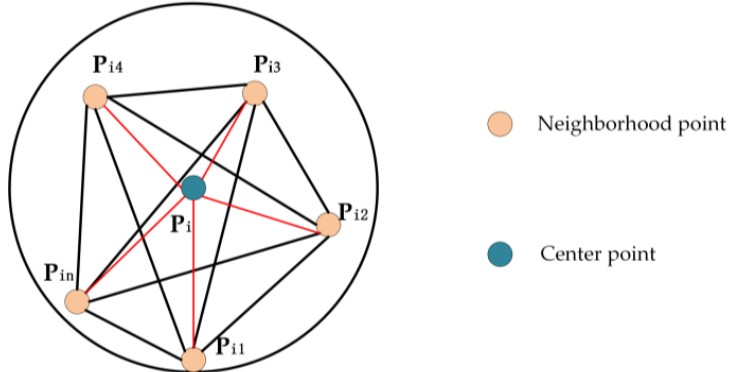

**Figure 3.** The local fully connected graph. $P_i$ is the center point, $P_{i1}$, $P_{i2}$ … $P_{in}$ is the points within the neighborhood.

**Point feature enhancement:** In order to make full use of the given feature information (RGB and high dimensional feature), the feature $r_i^k$ is then concatenated to the point feature $f_i^k$ to form the new enhanced feature $s_i^k$.

$$s_i^k = MLP\left(r_i^k \oplus f_i^k\right) \tag{4}$$

### 2.1.2. Attentive Pooling

Pooling operation can significantly reduce the number of model parameters and alleviate overfitting. At present, max pooling and average pooling, are the most commonly used methods; however, they easily lose some subtle but important features. With the development of the attention mechanism, related works have achieved good results by replacing the traditional pooling layer with the attention-pooling layer. Therefore, we use attentive pooling in this paper.

We use $S_i$ to describe the set of neighborhood of the enhanced features $s_i^k$ in the previous step, as shown in Equation (5).

$$S_i = \left\{ s_i^1, s_i^2, \ldots, s_i^{K-1}, s_i^K \right\} \tag{5}$$

where $k = 1,2,3 \dots K$.

We evaluate $s_i^k$ using a function $g(x)$ composed of an *Shared MLP* and a *softmax* function to obtain its attention weight $w_i^k$, as shown in Equation (6).

$$w_i^k = g\left(s_i^k, W\right) \tag{6}$$

where $W$ is the learnable weights of a *Shared MLP*.

Finally, we weight-sum each reinforcement feature $s_i^k$ and its corresponding weight $w_i^k$ to obtain $f_i'$, which is the result of attention pooling, as shown in Equation (7).

$$f_i' = \sum_{k=1}^{K} \left(w_i^k * s_i^k\right) \tag{7}$$

2.1.3. Dual Dilated Residual Aggregation Module

With the increase in the network layers, vanishing gradients and over-smoothing issues are likely to occur; thus, stacking multiple layers is not an effective way to improve performance. Moreover, with the receptive field increasing, repeatedly applying down-sampling after aggregation can cause spatial information loss, which reduces the accuracy of segmentation. Hence, inspired by DeepGCN [31], we introduced residual connections and dilated convolution into our network. In addition, to create a balance between efficiency and accuracy, we used two residual blocks; we call this dual dilated residual block (Figure 4).

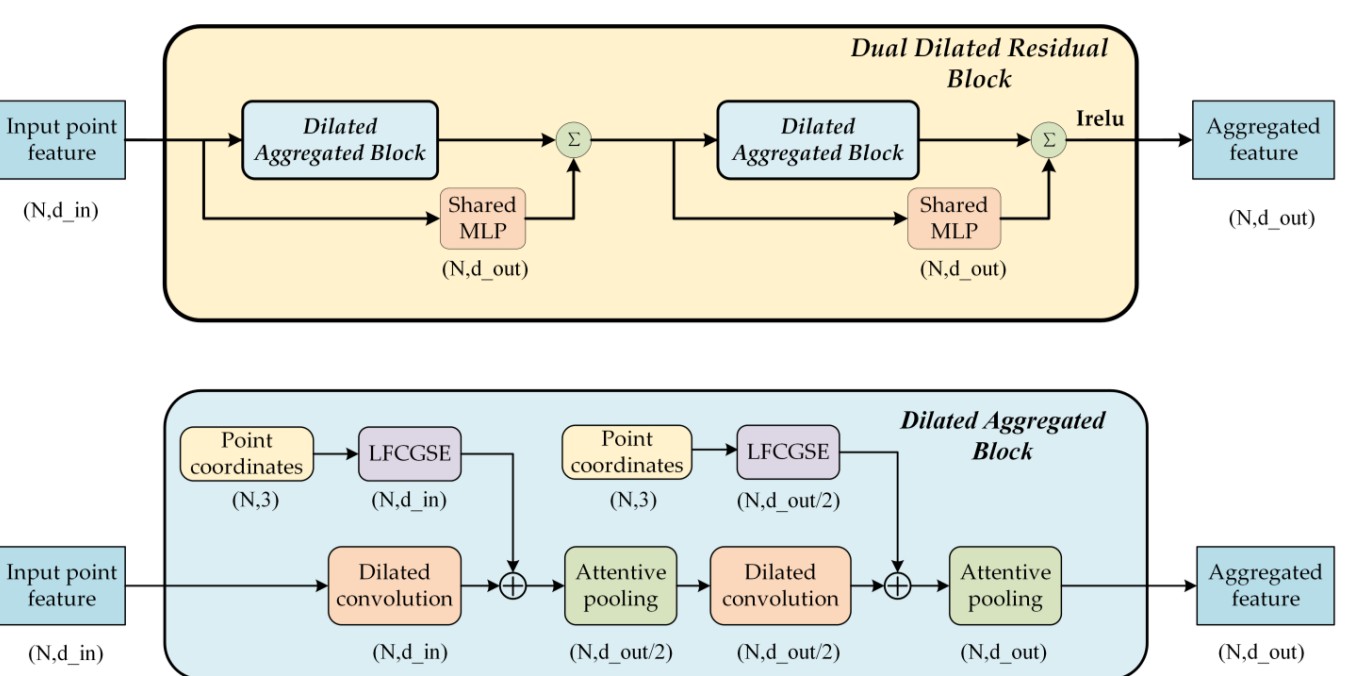

**Figure 4.** Dual dilated residual block.

*2.2. Instantiating Clustering*

To facilitate 3D reconstruction, the point set with the same semantic label needs to be separated into individual instances. Therefore, a clustering algorithm is necessary. There are already some clustering algorithms, such as k-mean and Euclidean clustering, but due to the existence of the adhesion in the segmentation, it is difficult to separate them through the existing clustering algorithm. Therefore, to minimize the impact of the adhesion, referring to the image erosion and dilation, we propose a morphology-based clustering algorithm for

point clouds, which uses erosion and dilation to group the points efficiently and accurately. As shown in Figure 5, our method of instantiating clustering includes the following steps:

(1) Point-cloud projection. The point cloud $P_l$ with the semantic label $l$ is projected onto a 2D plane to obtain the original projection point cloud $P_l^{2D}$, after which grid sampling is performed to obtain the sampled 2D point cloud $Q_l$.

(2) Point-cloud erosion. To remove the adhesive areas between two point clusters, the erosion operation is performed on the sampled point cloud. The points that do not have enough number of points in neighborhoods are removed.

(3) Euclidean clustering. Based on the corrosion on point clouds, the Euclidean clustering is used to obtain instantiate point sets $Q_l^{kernel} = \{Q_l^1, Q_l^2, Q_l^3, ...\}$, which is called projected point cloud kernel.

(4) Point-cloud dilation. The $Q_l^{kernel}$ is used as query point set, a dilation is performed on the original projected point cloud $P_l^{2D}$ to obtain the instantiating point set $P_l^i = \{P_l^1, P_l^2, P_l^3, ...\}$ (the indices of $P_l^{2D}$ are the same as those of $P_l$).

(5) Height recovery. The point-cloud height value is assigned from $P_l$ to $\{P_l^1, P_l^2, P_l^3, ...\}$, according to the indices.

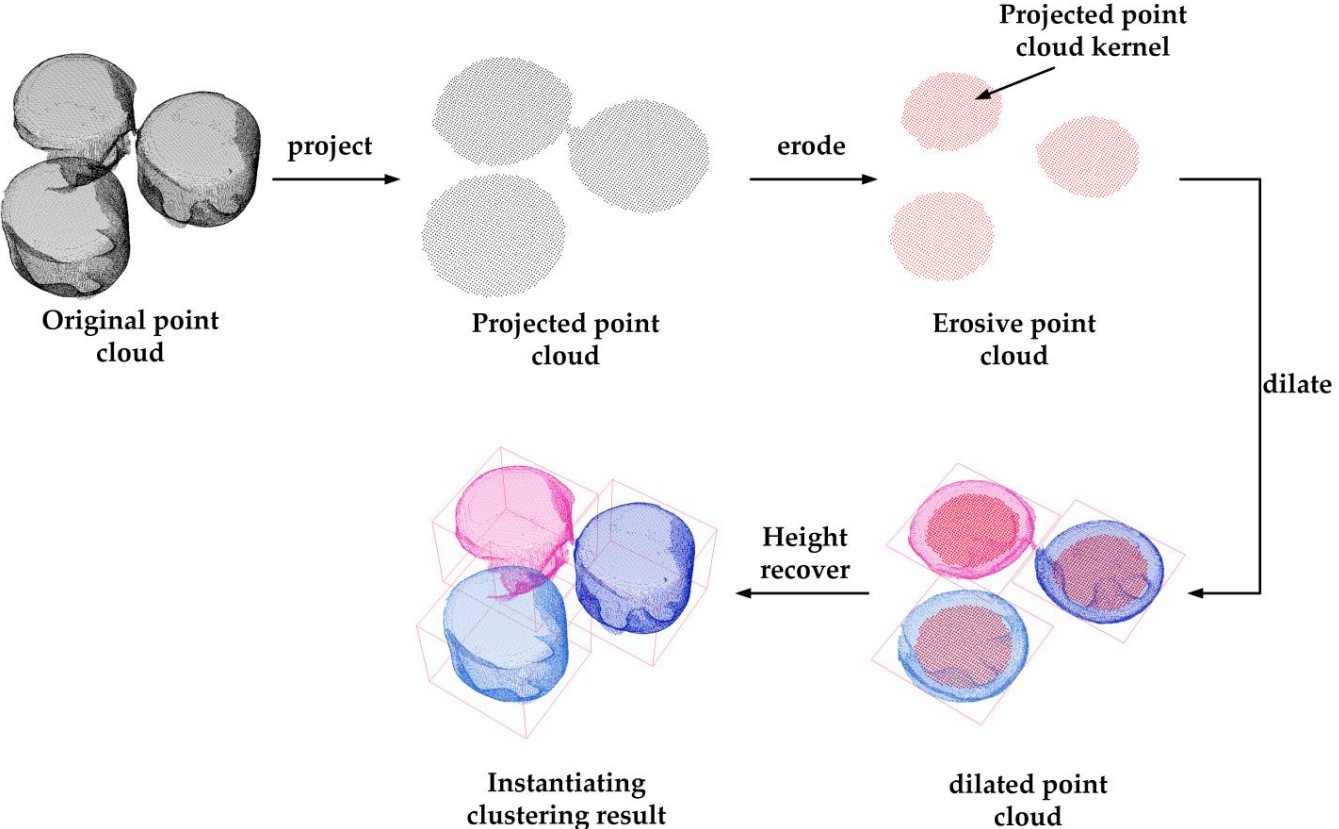

**Figure 5.** Flowchart of instantiating clustering.

### 2.3. Reconstruction of Unstructured Objects

Indoor unstructured objects generally refer to furniture, such as tables, chairs, beds and sofas. Compared with structured objects, such as walls and floors, the shape of the object is complex and difficult model directly by some sample geometric priors. Since the unstructured object is enumerable, the type of shape is limited, especially in some specific scenes, such as bedrooms, offices, parking, etc. Hence, to reconstruct the unstructured object rapidly, we built a 3D indoor model library and proposed a model-retrieval algorithm.

### 2.3.1. 3D-ESF Indoor Model Library

The 3D-ESF indoor model library consists of the 3D template model set and the ESF descriptor index.

(1)    3D template model set

Our 3D template model set is based on the ModelNet [32] datasets released by the Princeton Vision and Robotics Laboratory in 2015. The 3D model library contains 20 types of furniture, and each type contains hundreds of CAD (Computer Aided Design) models. A part of our template models is shown as Figure 6.

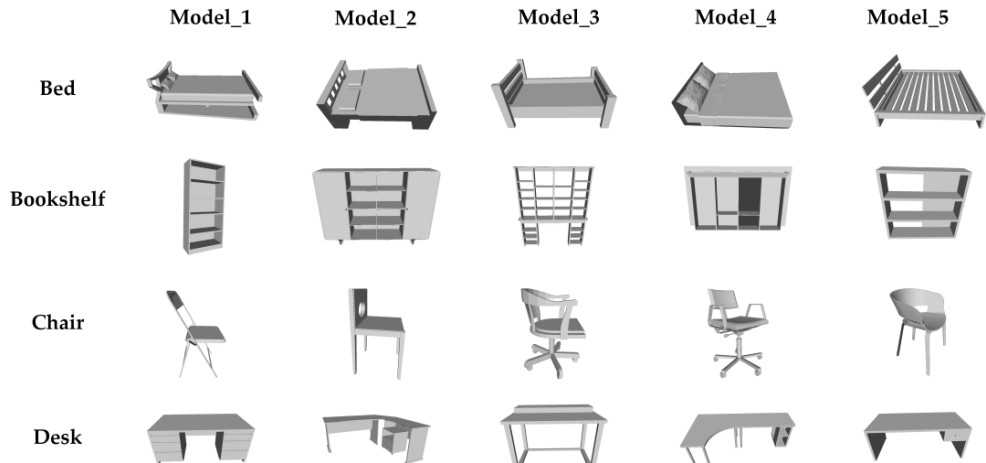

**Figure 6.** Several models of beds, bookcases, chairs, tables, etc. in the model library.

(2)    ESF descriptor index

The feature descriptors of point clouds are usually used for 3D object recognition. The ESF descriptor is a global 3D-shape descriptor proposed by Walter [27], which consists of 10 sets of 64-dimension shape-function histograms (640-dimension). They can be divided into 3 groups:

D2: The distances between three random points and distance ratio;

A3: The angles between three random points;

D3: The square root of the area of the triangle composed of three random points;

Note: All the measures are classified as IN (inside the model), OUT (outside the model), or MIXED (pass both the inside and outside of the model);

Uniform downsampling was performed on all of the models in the library, after which the ESF descriptors were calculated on the sampled point cloud and attached with the model as the index for retrieval.

### 2.3.2. Model Retrieval

With the semantic labels, the type of the model can be determined rapidly, but for each type, there are many variants with different shapes, such as office chairs, dining chairs, etc.; therefore, an efficient and accurate model-retrieval algorithm is essential. As shown in Figure 7, there are differences in the ESF descriptor histograms between the target point cloud and the template-model point cloud. Base on the ESF descriptors, we propose an ESF similarity-based model-retrieval algorithm to find the candidate models.

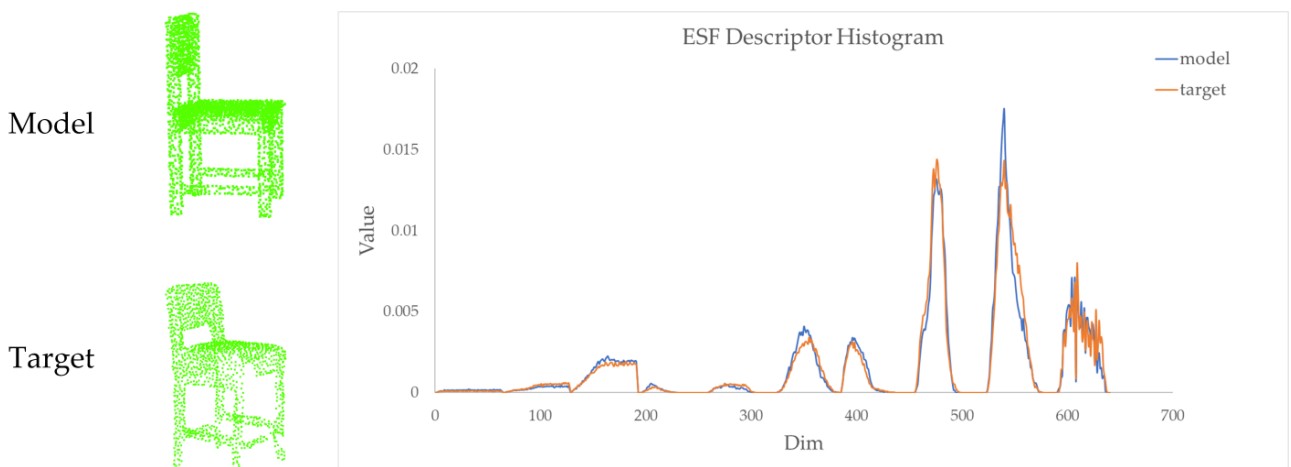

**Figure 7.** ESF descriptor histograms. The ESF descriptor histograms of target (orange) and model (blue).

The similarity between target point cloud and the template-model point cloud is calculated by using the weighted Euclidean distance of 640 dimension. Since the A3 has a greater impact than D2 and D3, to trade off the impact, the 640-dimension elements are divided into two parts ([1, 448], [449, 640]) and then calculated using the weighted Euclidean distance. The similarity is calculated as follows:

$$S(C_i, C_j) = w1 * \sum_{k=1}^{448} (C_{ik} - C_{jk})^2 + w2 * \sum_{k=449}^{640} (C_{ik} - C_{jk})^2 \tag{8}$$

where $C_i$ is the ESF-descriptor histogram of target point cloud, $C_j$ is the ESF-descriptor histogram of model point cloud, $w1$ and $w2$ are the weights of the first part and second part ($w1 = 0.8$, $w2 = 0.2$). $C_{ik}$ and $C_{jk}$ are the k-th dimension probability values of $C_i$ and $C_j$, respectively. According to the $S(C_i, C_j)$, the top $n$ candidates are preserved ($n = 3$).

### 2.3.3. Coarse-to-Fine Registration

To reconstruct the unstructured object in the scene, a coarse-to-fine registration method is proposed. The method registers the template model to the target point cloud while determining the most similar one from the retrieval candidates.

(1)    Projection-based coarse registration

Since some of the retrieved candidate models have tiny differences in shape, the coarse registration algorithm can be used to further determine the most similar model. In addition, in order to prevent the impact of point-cloud incompleteness and accelerate the registration, the target point cloud and sampling point cloud of candidate models are projected onto a 2D plane, after which the ISS (Intrinsic Shape Signatures) key point [33] and FPHF (Fast Point Feature Histograms) descriptor [34] combined registration method is used to evaluate the result, as follows:

Low-confidence point filtering. After the semantic segmentation, each point has a confidence; the point-cloud confidence heat map is shown in Figure 8, since the points with low confidence may lead to a large registration error, to improve the registration accuracy, the points with low confidence are filtered.

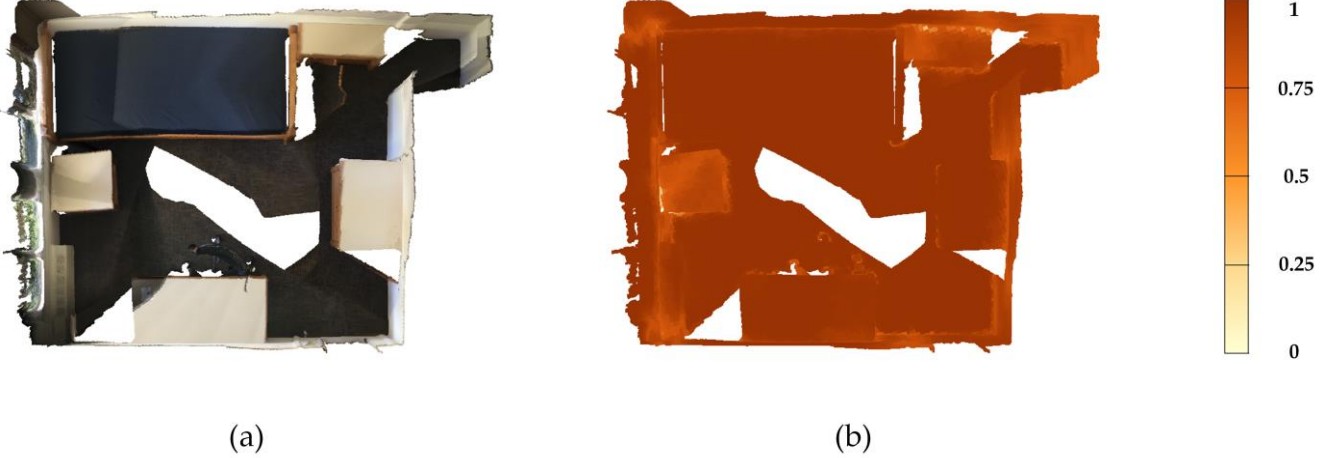

**Figure 8.** Low-confidence point filtering. (**a**) Original point cloud, (**b**) point-cloud confidence heat map after semantic segmentation.

(2) Scale adjustment. To adjust the scale of the template model to that of the target point cloud, we calculate the size ratios of their bounding boxes in the x,y,z direction. Considering the incomplete point cloud may cause the bounding box of the target point cloud to be shrink in a certain direction, the highest ratio in the x,y,z direction is selected as the scaling factor.

(3) Coarse registration. Based on the priors, the furniture is generally vertical and aligned with the floor. Hence, we project both the target point cloud and the template point cloud on the *xoy* plane. Next, ISS key point [33] detection is performed, which can improve the computational efficiency while maintaining the original features. Base on the ISS key points, the FPFH (Fast Point Feature Histograms) registration [34] is performed. In addition, the one with the lowest registration error among the candidates is selected.

(4) Model-to-scene fine registration

Based on the last step in the coarse registration, a fine registration is used to recover the precise position and orientation of the model in the scene. Instead of using the point-to-point registration method, to make full use of the precise normal of template model, we propose an inverse point-to-plane ICP (Iterative Closest Point) algorithm to precisely register the template model to the target point cloud (Equations (6)–(8)).

$$E(R,t) = \sum_{i=1}^{N} \left( (R(p_i) + t - q_i)^T n_i \right)^2 \tag{9}$$

$$R^*, t^* = argmin_{R,t} E(R,t) \tag{10}$$

$$R', t' = Inversion(R^*, t^*) \tag{11}$$

where $p_i$ is the *i*-th point in the target point cloud, $q_i$ is the *i*-th vertex on the template model, $n_i$ is the normal of $q_i$, R and T are the rotation and translation transformation. $R^*$ and $t^*$ are the optimal solution of $E(R,t)$. Since the template model should be registered to the target point cloud (plane-to-point), $(R', t')$ is the inverse transform of $(R^*, t^*)$.

### 2.4. Structured-Object Reconstruction

Structured objects, such as walls, floors, ceilings, etc., have regular plane structures. Therefore, we consider the reconstruction of the structured object as an outline and plane-extraction problem.

Floor and ceiling modeling. We model the floor and ceiling using a plane-fitting algorithm, such as RANSAC plane fitting, with the horizontal prior in Cartesian coordinate system.

Wall modeling. Due to the occlusion, most of the walls are difficult to extract with the plane-fitting algorithm. To perform a complete wall extraction, the vertical prior of wall is introduced. Therefore, we project the points of wall onto the floor plane, after which the RANSAC line-fitting algorithm is used to obtain the outline of the indoor scene.

## 3. Results

### 3.1. Point-Cloud Semantic-Segmentation Experiment

In order to demonstrate the efficacy of the LFCG-Net for semantic segmentation, we conducted our experiment over S3DIS [35] and ScanNet [36] datasets, and compared them with related methods. The experimental results verify the effectiveness and practicability of the proposed segmentation network.

To ensure fairness, we used the same parameters and training methods as [26]. We used an Adam optimizer, the batch size was 6, maximum epoch was 100, the number of input points was 40960 per batch, each point included (x, y, z, r, g, b, label), the ratio of the downsampling was 0.25, 0.25, 0.25, 0.25, and 0.5, and the initial learning rate was 0.01, which then decreased by 5% at each epoch. All the experiments were performed on NVIDIA RTX8000 GPU.

#### 3.1.1. The Semantic Segmentation on S3DIS

The S3DIS datasets was collected in six large indoor areas from three buildings with different architectural styles, including 11 room categories, 271 rooms, and more than 200 million points, which were composed of the spatial coordinates, colors and labels of 13 semantic categories. We used areas 1, 2, 3, 4 and 6 as the training sets and area 5 as the test set. Area 5 of the scene is shown in Figure 9.

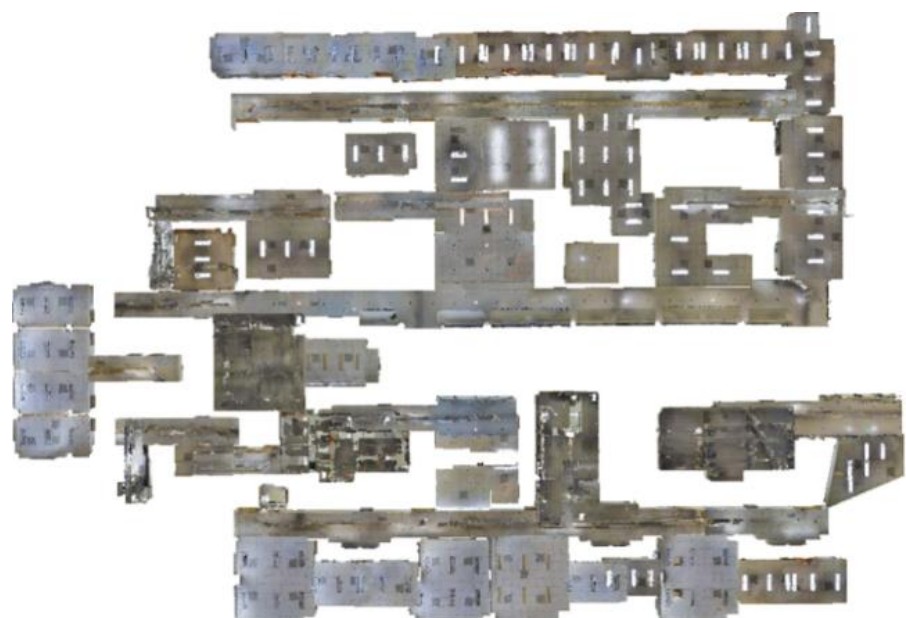

**Figure 9.** Area 5 of the S3DIS datasets.

The experimental results are shown in Table 1. This paper uses three widely applied metrics to test the semantic-segmentation performance of our network: Mean Intersection over Union (MIoU), Intersection over Union (IoU), and Overall Accuracy (OA). Compared to previous works, our LFCG-Net achieves significant improvement.

**Table 1.** Comparison of the semantic segmentation tested on the S3DIS area 5.

| Method | mIoU | mAcc | OA | ceiling | floor | wall | beam | column | window | door | table | chair | sofa | bookcase | board | clutter |
|---|---|---|---|---|---|---|---|---|---|---|---|---|---|---|---|---|
| PointNet [18] | 41.1 | 49.0 | - | 88.8 | 97.3 | 69.8 | 0.1 | 3.9 | 46.3 | 10.8 | 59.0 | 52.6 | 5.9 | 40.3 | 26.4 | 33.2 |
| PointCNN [21] | 57.3 | 63.9 | 85.9 | 92.3 | 98.2 | 79.4 | 0.0 | 17.6 | 22.8 | 62.1 | 74.4 | 80.6 | 31.7 | 66.7 | 62.1 | 56.7 |
| SPGraph [25] | 58.0 | 66.5 | 86.4 | 89.4 | 96.9 | 78.1 | 0.0 | 42.8 | 48.9 | 61.6 | 84.7 | 75.4 | 69.8 | 52.6 | 2.1 | 52.2 |
| PointWeb [37] | 60.3 | 66.6 | 87.0 | 92.0 | 98.5 | 79.4 | 0.0 | 21.1 | 59.7 | 34.8 | 76.3 | 88.3 | 46.9 | 69.3 | 64.9 | 52.5 |
| RandLA-Net [26] | 62.5 | 71.5 | 87.2 | 91.1 | 95.6 | 80.2 | 0.0 | 25.0 | 62.1 | 47.3 | 76.0 | 83.5 | 61.2 | 70.9 | 65.5 | 53.9 |
| MinkowskiNet [38] | 65.4 | 71.7 | - | 91.8 | 98.7 | 86.2 | 0.0 | 34.1 | 48.9 | 62.4 | 81.6 | 89.8 | 47.2 | 74.9 | 74.4 | 58.6 |
| **Ours** | **65.4** | **73.6** | **88.8** | **93.2** | **97.9** | **82.8** | **0.0** | **24.9** | **65.1** | **59.6** | **78.0** | **88.3** | **67.7** | **71.1** | **65.5** | **55.6** |

As shown in Figure 10, compared to the RandLA-Net method [26], our network can achieve better accuracy in many respects.

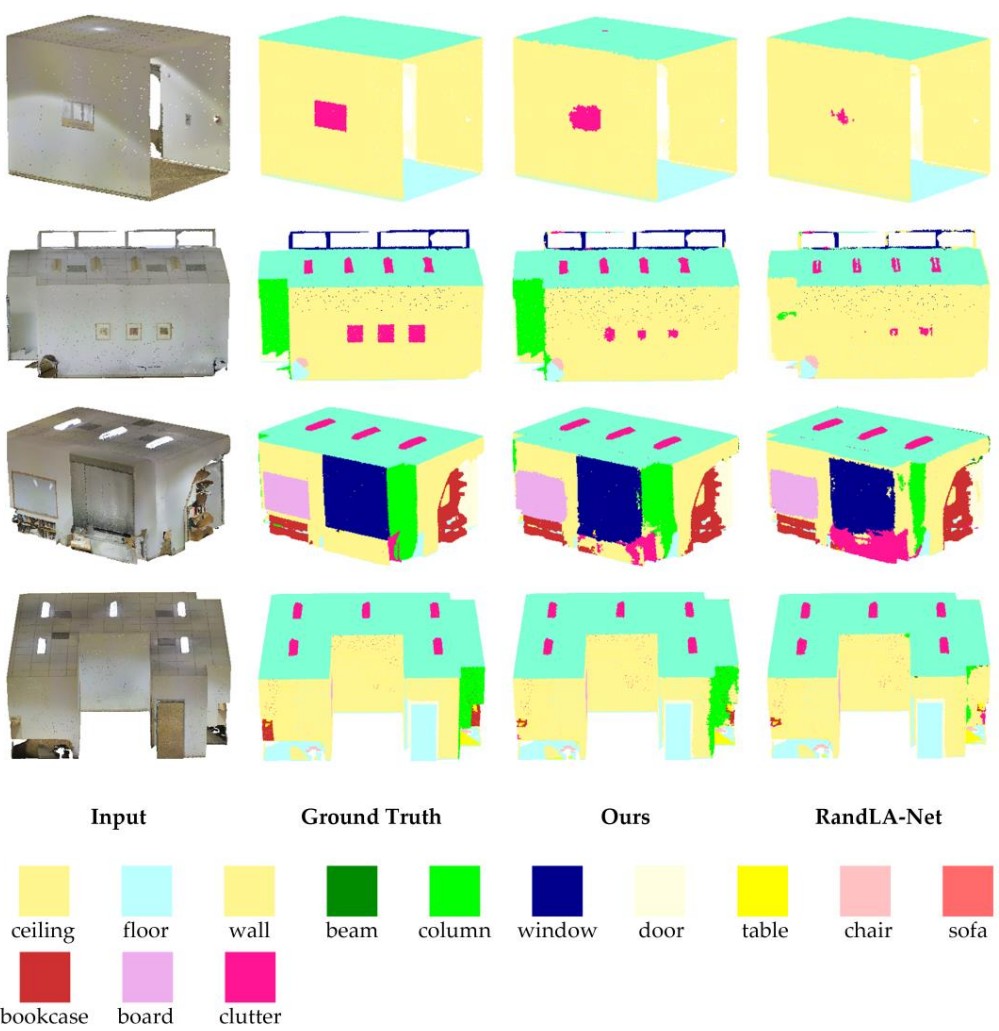

**Figure 10.** Visualization of semantic-segmentation results on S3DIS datasets area 5.

### 3.1.2. Semantic Segmentation on ScanNet

The ScanNet dataset is an RGB-D video dataset containing scene information from more than 1500 scans with annotations for 3D camera pose, surface reconstruction and instance-level semantic segmentation. Some scenes are shown in Figure 11.

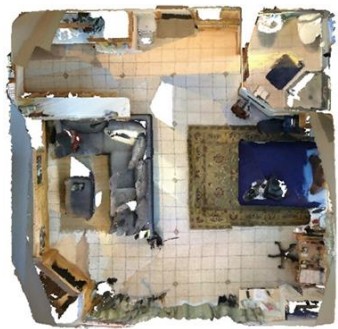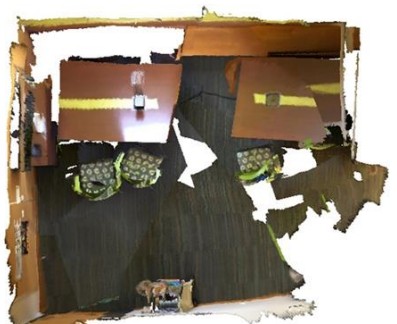

**Figure 11.** The ScanNet dataset.

The settings were similar to those in PointNet++ [19]. The last 1201 scenes of the total of 1503 were used as the training set and the remaining 302 scenes were used as the test set.

Table 2 shows the comparison with related methods on the ScanNet datasets. Our LFCG-Net achieved the best result, of 64.1% mIoU. It offers improvements of 2.2~30.2%.

**Table 2.** Comparison of the semantic segmentation on ScanNet datasets.

| Method | mIoU | wall | floor | chair | tabel | desk | bed | bookshelf | sofa | sink | bathtub | toilet | curtain | counter | door | window | Shower curtain | refrigerator | picture | cabinet | Other furniture |
|---|---|---|---|---|---|---|---|---|---|---|---|---|---|---|---|---|---|---|---|---|---|
| Pointnet++ [19] | 33.9 | 52.3 | 67.7 | 36.0 | 23.2 | 27.8 | 47.8 | 45.8 | 34.6 | 36.4 | 58.4 | 54.8 | 24.7 | 25.0 | 26.1 | 25.2 | 14.5 | 21.2 | 11.7 | 25.6 | 18.3 |
| RandLA-Net [26] | 61.9 | 70.7 | 95.5 | 80.4 | 65.2 | 61.0 | 85.1 | 76.7 | 74.8 | 61.8 | 81.7 | 76.4 | 58.0 | 58.6 | 38.6 | 43.7 | 63.5 | 47.1 | 27.8 | 29.9 | 41.5 |
| **Ours** | **64.1** | **73.2** | **95.4** | **83.0** | **66.1** | **60.8** | **86.1** | **76.8** | **77.8** | **64.3** | **80.7** | **77.4** | **58.9** | **56.2** | **45.4** | **44.2** | **71.5** | **51.8** | **28.8** | **35.7** | **46.7** |

### 3.1.3. Ablation Experiment

To further evaluate the effects of the fully connected graph spatial-encoding module and dual dilated residual block, we conducted the following ablation studies. All the ablation networks were trained on the S3DIS datasets for areas 1, 2, 3, 4 and 6 and tested on area 5. The ablation results are shown in Table 3.

**Table 3.** Ablation results.

| Operator | mIoU (%) |
|---|---|
| DRB and LocSE | 62.5 |
| DRB and LFCGSE | 64.3 |
| DDRB and LocSE | 64.4 |
| DDRB and LFCGSE | 65.4 |

(1) The local fully connected graph spatial encoding module (LFCGSE) is replaced with the local spatial encoding module (LocSE) in RandLA-Net [26], and the dual dilated residual block (DDRB) is replaced with the dilated residual block (DRB) in RandLA-Net [26].

(2) The dual dilated residual block (DDRB) is replaced with the dilated residual block (DRB) in RandLA-Net [26].

(3) The local fully connected graph spatial-encoding module (LFCGSE) is replaced with the local spatial-encoding module (LocSE) in RandLA-Net [26].

(4) The local fully connected graph spatial-encoding module (LFCGSE) and the dual dilated residual block (DDRB) are used.

### 3.2. Semantic Reconstruction Experimentation

To demonstrate the efficacy of the reconstruction of the unstructured objects, we also conducted a reconstruction experiment based on our 3D-ESF indoor-model library.

### 3.2.1. Unstructured-Object Reconstruction

We took chair retrieval as an example. Figure 12 shows the ESF feature histogram of the target point cloud and the sampled point cloud from template model.

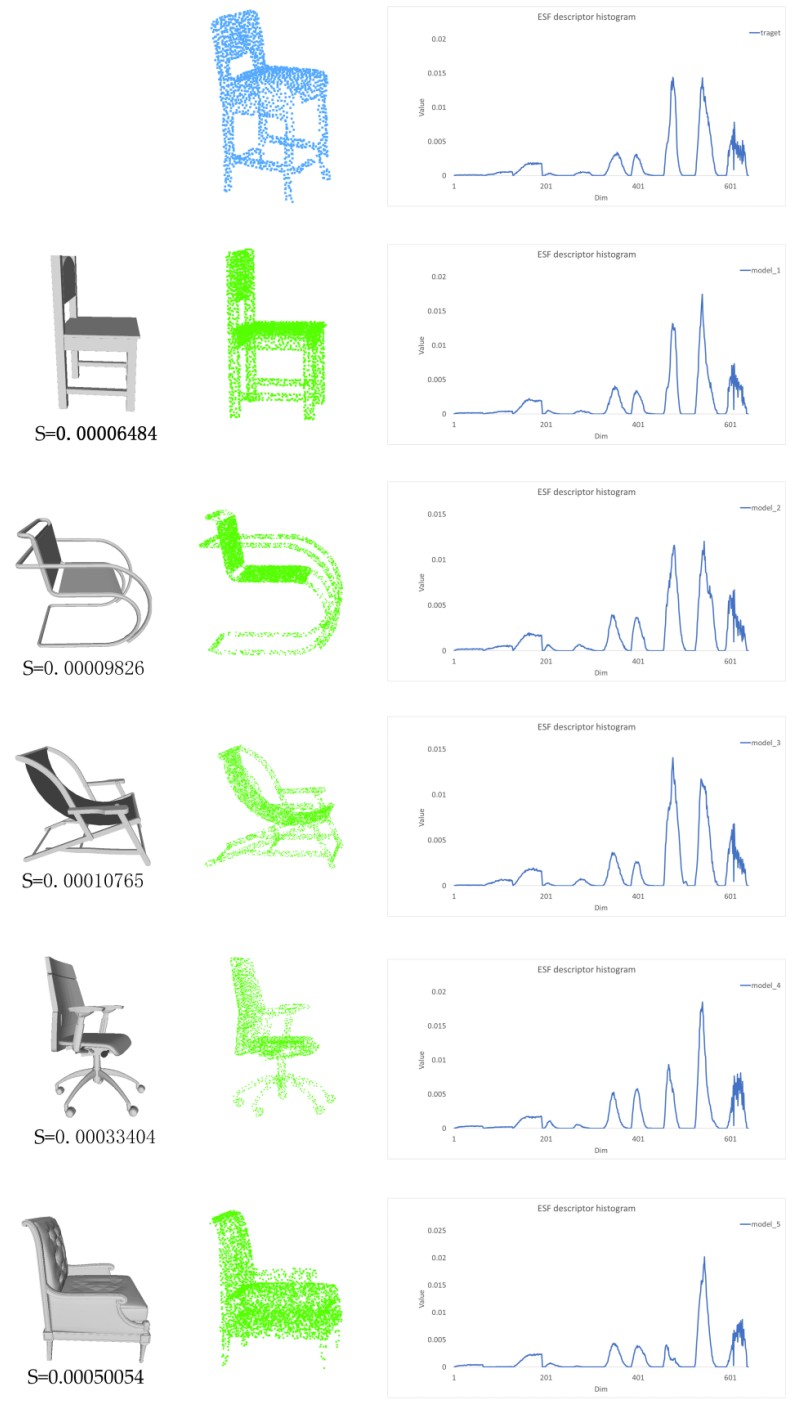

**Figure 12.** The first row (blue) is the target point cloud (sampling density is 0.02 m) and its ESF-descriptor histogram. The remaining rows are 5 candidate chair models, sampled point clouds and the ESF-descriptor histogram. The rows are sorted by the similarity distance S from target point cloud (lower is better).

By measuring the similarity S, the first three models are preserved. The results in Figure 12 show that the model with lower-similarity distance is more similar to the shape of the target point cloud.

Figure 13 shows the projection-based coarse registration. The one with lowest error has the most similar shape and the best registration result. With the ISS key point and FPFH descriptor combined-registration method, the registration time is reasonable, which offers benefits for large-scale scene reconstruction.

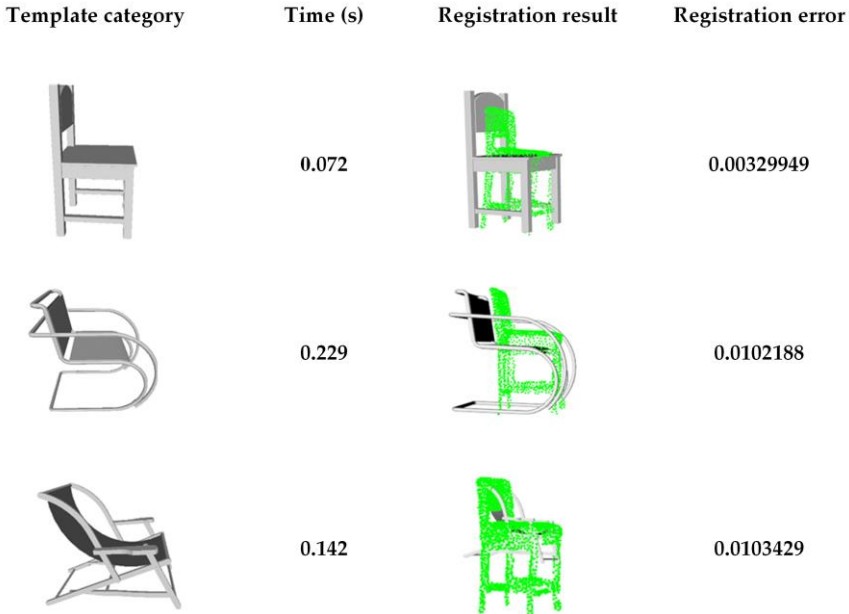

| Template category | Time (s) | Registration result | Registration error |
| --- | --- | --- | --- |
| | 0.072 | | 0.00329949 |
| | 0.229 | | 0.0102188 |
| | 0.142 | | 0.0103429 |

**Figure 13.** The following coarse registration experiment.

Figure 14 shows the fine registration results. Based on our inverse point-to-plane ICP registration, we achieved a better result within a reasonable time.

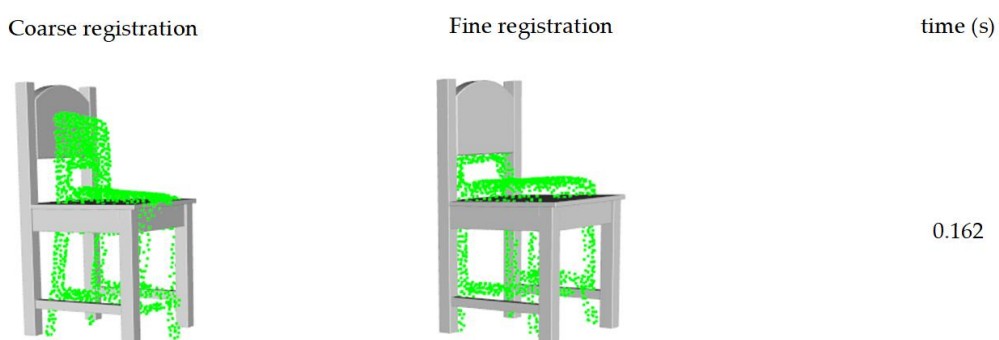

| Coarse registration | Fine registration | time (s) |
| --- | --- | --- |
| | | 0.162 |

**Figure 14.** Model-to-scene fine registration.

### 3.2.2. Comparison of Related Methods

To verify the advantages of our semantics-guided 3D reconstruction method, we also conducted the following comparison experiments. The experimental scene was from the ScanNet datasets. In the scene, there were 10 chairs, some of which were incomplete in the point cloud and which closely resembled the real environment. The point cloud was uniformly downsampled by 0.02 m.

Figure 15 is the comparison of the different registration methods. We compared the different registration methods with or without semantic information.

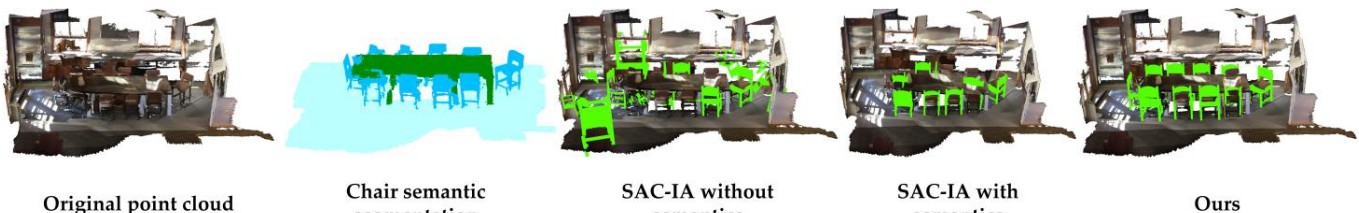

Original point cloud          Chair semantic          SAC-IA without          SAC-IA with          Ours
                              segmentation            semantics               semantics

**Figure 15.** Comparison of different registration methods.

(1)    SAC-IA (Sample Consensus Initial Aligment) without semantics. Performance of the SAC-IA registration;
(2)    SAC-IA with semantics. Performance of the SAC-IA registration on the point cloud with the same semantic label;
(3)    ISS+FPFH with semantics (ours).

*3.3. Overall Reconstruction*

By integrating the unstructured- and structured-object reconstruction, the whole indoor scene was reconstructed completely. The final result is shown as Figure 16.

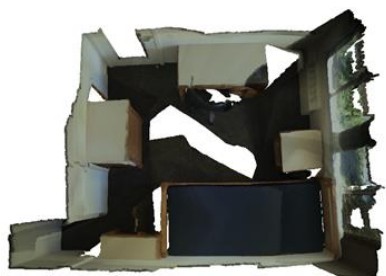 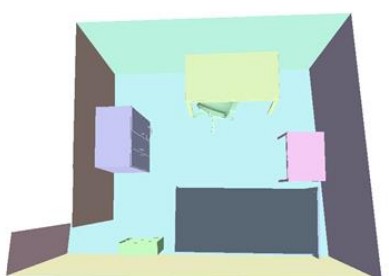 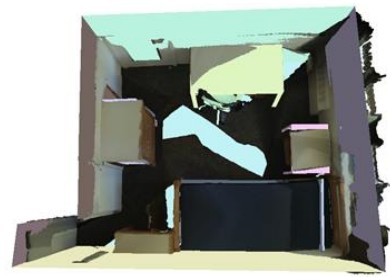

**Figure 16.** Overall reconstruction: (**left**) the original input point cloud, (**middle**) the reconstruction result, (**right**) the overlay of (**left**) and (**middle**).

From Figure 16, we can see that our method can reconstruct indoor scenes, even if the scene has some holes on the wall and floor.

**4. Discussion**

It is shown through experimental studies that the point-cloud semantic-segmentation network LFCG-Net, proposed in this paper, outperforms other state-of-the-art networks on two different indoor-point-cloud datasets. Furthermore, the retrieval and registration algorithms based on the model library, proposed in this paper, can reconstruct the indoor scene well. Some phenomena noted in these experiments are worth discussing.

We built a point-cloud semantics-and-primitives-guided pipeline for indoor 3D reconstruction. As shown in our experiment, the point-cloud semantic-segmentation network proposed in this paper, LFCG-Net, achieves the best results on two mainstream datasets. Based on the 3D-ESF indoor-model library, the proposed semantic-reconstruction method is able to reconstruct unstructured objects and structured objects from unordered point clouds, even if the scanning data are incomplete. The method demonstrates the use of the combination of deep learning and traditional methods in 3D scene reconstruction.

In the semantic-segmentation experiment, the proposed LFCG-Net achieved a significant improvement, of 2.2~30.2%, taking SanNet as an instance. This mainly benefits from the local fully connected encoding module (LFCGSE) and the dual dilated residual block module (DDRB). Our ablation experiment shows that the LFCGSE module can provide more effective encoding. Compared with the existing method, it can obtain a complete description of the context–shape relation, which is important for improving accuracy.

Furthermore, without the DDRB, the network cannot achieve the expected accuracy. Over-smoothing and gradient disappearance are the main factors that affect the performance when training the deep gnn. To solve this problem, in the DDRB module, the graph dilated residual convolution was applied, which suppressed the occurrence of the over-smoothing and gradient disappearance.

In the semantic 3D reconstruction experiment, based on the semantic labels from the semantic segmentation, we applied a primitive-based-reconstruction method, which used the 3D-ESF indoor library and the coarse-to-fine registration algorithm to rapidly reconstruct the 3D scene from the incomplete and noisy point cloud. The primary difference between the indoor scene and the outdoor scene is that the indoor object is enumerable. Based on this, we built the 3D-ESF indoor-model library. The most similar model can be retrieved by using the ESF descriptors and the point-cloud semantic labels. Compared with the surface-based reconstruction method, the proposed method can provide complete 3D models, even if there is some incompleteness and noise in the point cloud. To register the model into the scene accurately, we propose the coarse-to-fine registration algorithm. As shown in the comparison of related work, with the guidance of point-cloud semantic labels, our algorithm achieves better accuracy, mainly because the guidance of semantics provides an effective local space for registration instead of global space, which significantly protects our registration algorithm from falling into the local optimum.

## 5. Conclusions

In this paper, we proposed a semantics-and-primitive-guided indoor 3D reconstruction method, which can reconstruct indoor scenes from incomplete and noisy point clouds. The proposed LFCG-Net and semantic reconstruction mechanisms can make full use of semantic information to recover 3D indoor scenes completely. LFCG-Net mainly includes two key modules, LFCGSE and DDRB. The effects of the LFCGSE and DDRB modules were verified by ablation studies on S3DIS area 5. Compared with the state-of-the-art point-cloud semantic-segmentation models on the public Scannet and S3DIS datasets, the results showed that LFCG-Net offers a significant improvement in the overall segmentation accuracy. The proposed indoor 3D reconstruction method is based on the semantic-segmentation results and the 3D-ESF indoor-model library. The method retrieves the most similar models in the library with the ESF descriptors and semantic labels and then, with the guidance of point-cloud semantics, the model retrieved is registered by a coarse-to-fine registration algorithm. The experiments show that our approach has better practicality, robustness and accuracy.

**Author Contributions:** Conceptualization, Q.W., H.A. and L.Z.; methodology, Q.W. and T.W.; software, T.W.; validation, T.W.; formal analysis, Q.W. and T.W.; writing—original draft preparation, T.W.; writing—review and editing, Q.W.; project administration, L.Z. and H.A.; funding acquisition, L.Z. and H.A.; All authors have read and agreed to the published version of the manuscript.

**Funding:** This research was funded by the National Key R & D Program of China, grant number 2019YFB1405600 and Fundamental Research Funds for Chinese Academy of Surveying and Mapping, grant numbers AR2201 and AR2209.

**Data Availability Statement:** The data supporting reported results are available in Stanford 2D-3D-Semantics Dataset [35] and ScanNet Benchmark Challenge [36] (https://shapenet.cs.stanford.edu/media/indoor3d_sem_seg_hdf5_data.zip, http://www.scan-net.org).

**Acknowledgments:** The authors thank the anonymous reviewers very much for their valuable comments, which greatly improved this paper.

**Conflicts of Interest:** The authors declare no conflict of interest.

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
