# Peer review of "Semantics-and-Primitives-Guided Indoor 3D Reconstruction from Point Clouds"

_remotesensing, doi:10.3390/rs14194820_

Round 1

Reviewer 1 Report

In this paper, the authors propose a semantics guided indoor 3D reconstruction method, which can reconstruct indoor scene from incomplete and noise point cloud. This method is based on LFCG-Net (local fully connected graph network) and semantic reconstruction mechanisms that can use semantic information to recover the 3D indoor scene in a much more efficient way than existing methods. The theoretical background of the applied method is presented in a clear and consistent manner. The selected application examples and the obtained results of the conducted experiments show that the proposed method has better accuracy and robustness to the incompleteness and noise of point cloud than the methods used so far. Therefore, the paper contains a satisfactory scientific contribution that qualifies it for publication in a journal.

The following comments are addressed to the authors:

-         - The authors use a number of abbreviations in the text that are not explicitly explained (For example: mIoU - average value of intersection of union). Probably the authors assume that those abbreviations are well known to readers who deal with this scientific field. However, it would be good if abbreviations were given at the beginning of the text together with their full meaning when they are first mentioned, so that even readers who are not experts in this scientific field could read the paper more easily.

-          - The authors used the Adam optimizer for LFCG-Net training, where the batch size is set as 6. The authors should explain why they chose that particular value. Could increasing the batch size value increase the efficiency of network training?

-          - It is not entirely clear whether the authors used the batch normalization technique during network training. Perhaps this implementation could increase the accuracy of the applied procedure.

Author Response

Dear Editors and Reviewers:

Thank you for your letter and for the reviewers’ comments concerning our manuscript entitled “Semantics guided indoor 3D reconstruction” (Manuscript ID:remotesensing-1854179). Those comments are all valuable and very helpful for revising and improving our paper, as well as the important guiding significance to our researches. We have studied comments carefully and have made correction which we hope meet with approval. Every revisions to the manuscript have been marked up using the “Track Changes” function in the paper. The main corrections in the paper and the responds to the reviewer’s comments are as follows:

Responds to the reviewer’s comments:

Reviewer #1:

1.The authors use a number of abbreviations in the text that are not explicitly explained (For example: mIoU - average value of intersection of union). Probably the authors assume that those abbreviations are well known to readers who deal with this scientific field. However, it would be good if abbreviations were given at the beginning of the text together with their full meaning when they are first mentioned, so that even readers who are not experts in this scientific field could read the paper more easily.

Response: I am sorry that this is what I overlooked, now I've added the full meaning of these abbreviations in the paper.

2.The authors used the Adam optimizer for LFCG-Net training, where the batch size is set as 6. The authors should explain why they chose that particular value. Could increasing the batch size value increase the efficiency of network training?

Response: Properly Increasing the batch size can improve the training efficiency, but the memory will also increase. Considering the memory capacity of the machine, And to ensure fairness, we use the same parameters and training methods as RandLA-Net.

We have added the explaination in the line 356.

3.It is not entirely clear whether the authors used the batch normalization technique during network training. Perhaps this implementation could increase the accuracy of the applied procedure.

Response: We have seriously considered your suggestion. We have used dropout in the network to alleviate the occurrence of overfitting. The simultaneous use of batch normalization and dropout will make the effect worse. So we do not use batch normalization.

Reviewer 2 Report

Dear Authors,

The subject of the study is interesting and topical, with scientific and practical importance.

The introduction is presented correctly, in accordance with the subject. Numerous scientific articles, in concordance to the topic of the study, were consulted.

Methodology of the study was clearly presented, and appropriate to the proposed objectives.

The obtained results are important and have been analyzed and interpreted correctly, in accordance with the current methodology.

Some suggestions were made in the article.

The following aspects are brought to the attention of the authors.

1.

Abstract

The Abstract could be revised to be more consistent with the topic covered.

It is recommended to consider the recommendations in the Instructions for Authors, Remote Sensing journal.

2.

Text settings

It is recommended that you consider the recommendations in the Instructions for Authors, and Microsoft Word template, the Remote Sensing journal, regarding to text settings.

Styles: MDPI_3.1_text

3.

Space before citing a bibliographic source in the text

eg

Page 2, row 76

DCNN [15]” instead of “DCNN[15]

Several suggestions were made in the article

4.

Table presentation

It is recommended to analyze whether tables 4 and 5, page 16 would not be more suitable to be presented as figures instead of tables.

5.

The Discussions is presented integrated in the Results chapter

Perhaps a more consistent reference to other bibliographic sources would be appropriate in this case.

6.

Conclusions

The chapter could better present / highlight the strengths of this study.

7.

Author Contributions:

Needs revision in accordance with Instructions for Authors, and Microsoft Word template, Remote Sensing journal

eg

The following statements should be used “Conceptualization, X.X. and Y.Y.; methodology, X.X.;

8.

References

The References chapter needs to be revised, according to the Instructions for Authors, Remote Sensing journal.

Author 1, A.B.; Author 2, C.D. Title of the article. Abbreviated Journal Name Year, Volume, page range.

Eg

“Wang, C.; Cho, Y.K.; Changwan, K. Automatic BIM component extraction from point clouds of existing buildings for sustainability applications. Autom. Constr. 2015, 56, 1–13. https://doi.org/10.1016/j.autcon.2015.04.001.”

Instead of

“Wang, Chao, Yong K. Cho, and Changwan Kim. 2015. “Automatic BIM Component Extraction from Point Clouds of Existing Buildings for Sustainability Applications.” Automation in Construction 56 (August): 1–13. https://doi.org/10.1016/j.autcon.2015.04.001.”

Author Response

Dear Editors and Reviewers:

Thank you for your letter and for the reviewers’ comments concerning our manuscript entitled “Semantics guided indoor 3D reconstruction” (Manuscript ID: remotesensing-1854179). Those comments are all valuable and very helpful for revising and improving our paper, as well as the important guiding significance to our researches. We have studied comments carefully and have made correction which we hope meet with approval. Every revisions to the manuscript have been marked up using the “Track Changes” function in the paper. The main corrections in the paper and the responds to the reviewer’s comments are as follows:

Responds to the reviewer’s comments:

Reviewer #2:

  1. Abstract

The Abstract could be revised to be more consistent with the topic covered.

It is recommended to consider the recommendations in the Instructions for Authors, Remote Sensing journal.

Response: We have revised the Abstract and used more precise descriptions, such as “The result indicate that…” to make it consistent with our theme.

  1. Text settings

It is recommended that you consider the recommendations in the Instructions for Authors, and Microsoft Word template, the Remote Sensing journal, regarding to text settings.

Response: We have modified the text settings as required by the Microsoft Word template.

  1. Space before citing a bibliographic source in the text

eg

Page 2, row 76

“DCNN [15]” instead of “DCNN[15]”

Several suggestions were made in the article

Response: We've added spaces before before citing a bibliographic source in the text.

  1. Table presentation

It is recommended to analyze whether tables 4 and 5, page 16 would not be more suitable to be presented as figures instead of tables.

Response: Thank you very much for your suggestion.We have replaced tables 4 and 5 by figure 14 and 15 respectively.

  1. The Discussions is presented integrated in the Results chapter

Perhaps a more consistent reference to other bibliographic sources would be appropriate in this case.

Response: We've split the discussion section out of the result and discussed it in more detail

  1. Conclusions

The chapter could better present / highlight the strengths of this study.

Response: We have revised the conclusions to be more detailed and explicit to highlight the strengths of this study.

  1. Author Contributions:

Needs revision in accordance with Instructions for Authors, and Microsoft Word template, Remote Sensing journal

eg

“The following statements should be used “Conceptualization, X.X. and Y.Y.; methodology, X.X.;”

Response: I'm sorry we neglected these. We have revised author contributions, according to Instructions for Authors and Microsoft Word template.

  1. References

The References chapter needs to be revised, according to the Instructions for Authors, Remote Sensing journal.

“Author 1, A.B.; Author 2, C.D. Title of the article. Abbreviated Journal Name Year, Volume, page range.”

Eg

“Wang, C.; Cho, Y.K.; Changwan, K. Automatic BIM component extraction from point clouds of existing buildings for sustainability applications. Autom. Constr. 2015, 56, 1–13. https://doi.org/10.1016/j.autcon.2015.04.001.”

Instead of

“Wang, Chao, Yong K. Cho, and Changwan Kim. 2015. “Automatic BIM Component Extraction from Point Clouds of Existing Buildings for Sustainability Applications.” Automation in Construction 56 (August): 1–13. https://doi.org/10.1016/j.autcon.2015.04.001.”

Response: I'm sorry we got the citation format wrong. We have revised the citation, according to Instructions for Authors and Microsoft Word template. And the citation format we used is MDPI ACS Style v3.

Reviewer 3 Report

Comments to article: Semantics guided indoor 3D reconstruction

Line 2: The title is too short and generic and could be more specific regarding how it has been done in the presented work and the algorithm used.

Lines 7 to 17: The summary is correct, but it must be structured by clearly introducing terms such as “the objective of this work is…”, “for them the methodology followed is…”. “The results indicate that…”, “therefore it is concluded that…”. Also at the beginning it should be indicated why or why it is important to understand the 3D scene.

Lines 20 to 100: The introduction in these lines, although some more reference could be added to those mentioned, are understood to be correct. They should explain (at least once) what some acronyms mean or introduce a list at the beginning of the work. Regarding the drawbacks of the current methods, they are generalized in excess since some of the mentioned methods have already overcome some of these drawbacks.

Lines 101 to 106: The proposal is interesting, although it could have been worded more clearly in some respects.

Lines 107 to 320: Regarding the general description of the methodology, it is interesting and although it appears to be quite exhaustive, it does not reveal some details that would facilitate its reproducibility, especially in relation to the LFCG-Net.

The architecture of the neural network is supposed to be correct, although we reiterate that it is an assumption because with the information provided it is difficult to reproduce for verification purposes, at least in terms of the most practical details for the purposes of what is indicated. .

There are aspects, such as pooling instances, that are also unclear as to how they can be done in practice for reproducibility purposes.

It is necessary to clarify the question of unstructured objects and their relationship with the place and if this can be a factor of interest.

The comparison of histograms is interesting in order to find similarities by means of the weighted Euclidean distance and, in this sense, the use of partitions is of interest in some cases, since it may be that the similarities are great in a part of the histogram and with this it is enough for identification.

In general, it is correct, but it is insisted that in many aspects an explanation and sufficient data are not offered to be able to reproduce the methodology in a more or less simple and practical way.

Lines 321 to 423: regarding the results, you are missing some explanations of justification for use, choice of some elements, parameters and values ​​(Adam optimizer, batch size, maximum epoch, number of entry points per batch, etc. .).

In general, this section is extensive in terms of comparison with others and testing and verification, but a statistical study of "accurasy", "precision", "recall", error, loss, etc., in general and by class, is lacking. .

Lines 424 to 433: regarding the conclusions, they are too brief.

The semantic-guided 3D interior reconstruction method is understood to have limitations that have not been exposed, since it is professed that an interior scene cannot always be reconstructed from an incomplete point cloud and noise.

The proposed LFCG-Net has not yet been developed in such a way that it is easily reproducible, and the semantic reconstruction mechanisms cannot make full use of the semantic information to fully recover the 3D interior scene. Although it is recognized that, although the experiments show robustness and precision and the semantic information is used as prior information to incorporate it in the reconstruction process, there are some ways to obtain stronger prior information that should be indicated.

Author Response

Dear Editors and Reviewers:

Thank you for your letter and for the reviewers’ comments concerning our manuscript entitled “Semantics guided indoor 3D reconstruction” (Manuscript ID:remotesensing-1854179). Those comments are all valuable and very helpful for revising and improving our paper, as well as the important guiding significance to our researches. We have studied comments carefully and have made correction which we hope meet with approval. Every revisions to the manuscript have been marked up using the “Track Changes” function in the paper. The main corrections in the paper and the responds to the reviewer’s comments are as follows:

Responds to the reviewer’s comments:

Reviewer #3:

  1. Line 2: The title is too short and generic and could be more specific regarding how it has been done in the presented work and the algorithm used.

Response: Our title has been revised to “Semantics and primitives guided indoor 3D reconstruction from point clouds”, to make it more specific and clear.

  1. Lines 7 to 17: The summary is correct, but it must be structured by clearly introducing terms such as “the objective of this work is…”, “for them the methodology followed is…”. “The results indicate that…”, “therefore it is concluded that…”. Also at the beginning it should be indicated why or why it is important to understand the 3D scene.

Response: Thanks for your suggestion, we have revised the introduction. We have added a description of the importance of the 3D scene and used relevant terminology to make the description of the introduction more formal and accurate.

  1. Lines 20 to 100: The introduction in these lines, although some more reference could be added to those mentioned, are understood to be correct. They should explain (at least once) what some acronyms mean or introduce a list at the beginning of the work. Regarding the drawbacks of the current methods, they are generalized in excess since some of the mentioned methods have already overcome some of these drawbacks.

Response: We are sorry for our negligence. We have added descriptions of some acronyms, such as "ESF(Ensemble of Shape Functions)". Regarding the shortcomings of the current method, we have used a more appropriate description to avoid generalizing it in excess.

  1. Lines 101 to 106: The proposal is interesting, although it could have been worded more clearly in some respects.

Response: We have revised it as suggested. We used clearer and more logical language to describe proposal.

  1. Lines 107 to 320: Regarding the general description of the methodology, it is interesting and although it appears to be quite exhaustive, it does not reveal some details that would facilitate its reproducibility, especially in relation to the LFCG-Net.

The architecture of the neural network is supposed to be correct, although we reiterate that it is an assumption because with the information provided it is difficult to reproduce for verification purposes, at least in terms of the most practical details for the purposes of what is indicated.

Response: We're sorry that we didn't provide sufficient details for the reproduction of the network. In the revision, we have provided a more specific description of the overall flow of the network and drawn the network architecture diagram more correctly. We have added the downsampling method and the number of sampling layers in the encoding process, the upsampling method in the decoding process, the specific process of pooling, the use of dropout, and the loss function used, to facilitate readers to reproduce the experiment.

There are aspects, such as pooling instances, that are also unclear as to how they can be done in practice for reproducibility purposes.

Response: We have added a detailed description and formulation of the pooling method in lines 194 to 208. The pooling method we used were attention pooling.

It is necessary to clarify the question of unstructured objects and their relationship with the place and if this can be a factor of interest.

Response: Unstructured objects mainly refer to objects that cannot be modeled by simple plane fitting methods, such as tables and chairs. Structured objects mainly refer to walls, beams, slabs and columns. Structured objects and unstructured objects are combined into the entire scene. For unstructured objects, such as chairs, after registering the chair point cloud with the model, we use coordinate transformation to convert the most similar chair model move in the scene. At this time, the position of the chair model is consistent with the position of the chair point cloud.

The comparison of histograms is interesting in order to find similarities by means of the weighted Euclidean distance and, in this sense, the use of partitions is of interest in some cases, since it may be that the similarities are great in a part of the histogram and with this it is enough for identification.

Response: Thanks for your affirmation.

In general, it is correct, but it is insisted that in many aspects an explanation and sufficient data are not offered to be able to reproduce the methodology in a more or less simple and practical way.

Response: Thanks for your suggestion. We've added some more detailed experimental details in the revised version to help others reproduce the experiment

6.Lines 321 to 423: regarding the results, you are missing some explanations of justification for use, choice of some elements, parameters and values (Adam optimizer, batch size, maximum epoch, number of entry points per batch, etc. .).

Response: To ensure fairness and facilitate comparison of experimental results, we use the same parameters and training methods as RandLA-Net, including optimizer, training batch, number of iterations, etc. We add an explanation on line 356.

In general, this section is extensive in terms of comparison with others and testing and verification, but a statistical study of "accuracy", "precision", "recall", error, loss, etc., in general and by class, is lacking. .

Response: Semantic segmentation experiments are quantitative. For comparison with other methods, we use mIoU (Mean Intersection over Union), IoU (Intersection over Union), OA (Overall Accuracy). As for the 3D reconstruction results, since our method is relatively novel, we use the model registration method for modeling, and there is no real value of the modeling for us to compare. So it is mainly qualitative analysis.

  1. Lines 424 to 433: regarding the conclusions, they are too brief.

Response: We have revised the conclusions to be more detailed and explicit to highlight the strengths of this study.

  1. The semantic-guided 3D interior reconstruction method is understood to have limitations that have not been exposed, since it is professed that an interior scene cannot always be reconstructed from an incomplete point cloud and noise.

Response: In fact, reconstructing 3D models from incomplete point clouds and noise has always been difficult, especially for whole indoor scenes. The first advantage of our method is the ability to combine semantic information to classify indoor scene point clouds and then model them; the second advantage is that the entire process is automated and does not require human involvement. The third advantage is that the reconstructed scene is very regular. Compared to other methods, our method has some degree of novelty and intelligence.

  1. The proposed LFCG-Net has not yet been developed in such a way that it is easily reproducible, and the semantic reconstruction mechanisms cannot make full use of the semantic information to fully recover the 3D interior scene. Although it is recognized that, although the experiments show robustness and precision and the semantic information is used as prior information to incorporate it in the reconstruction process, there are some ways to obtain stronger prior information that should be indicated.

Response: Thank you very much for your suggestion, in the revised version we have added more experimental details to the LFCGSE experiments to help readers reproduce. In addition, our semantic information reconstruction method can cover twenty main indoor objects, and the model library can also be expanded according to requirements. Indoor scenes can be automatically reconstructed as much as possible.

Round 2

Reviewer 3 Report

So, with this email I transmit the acceptance of the document to be published.